# Deconstructing Self-Bias in LLM-generated Translation Benchmarks

## Abstract

As large language models (LLMs) begin to saturate existing benchmarks, automated benchmark creation using LLMs (LLM-as-a-benchmark) has emerged as a scalable alternative to slow and costly human curation. While these generated test sets have to potential to cheaply rank models, we demonstrate a critical flaw. LLM-generated benchmarks systematically favor the model that created the benchmark: *they exhibit self-bias* on low resource languages to English translation tasks. We show three key findings on automatic benchmarking of LLMs for translation: First, this bias originates from two sources: the generated test data (LLM-as-a-testset) and the evaluation method (LLM-as-an-evaluator), with their combination amplifying the effect. Second, self-bias in LLM-as-a-benchmark is heavily influenced by the model's generation capabilities in the source language. For instance, we observe more pronounced bias in into-English translation, where the model's generation system is developed, than in out-of-English translation tasks. Third, we observe that low diversity in source text is one attribution to self-bias. Our results suggest that improving the diversity of these generated source texts can mitigate some of the observed self-bias.

## 1 Introduction

The rapid advancements in Large Language Models (LLMs) have led to an unprecedented saturation of existing, meticulously human-curated benchmarks. This phenomenon exposes two critical, intertwined challenges: traditional benchmark creation is too laborious and expensive to keep pace with rapid model development, and this challenge is compounded by the inherent difficulty of constructing high-quality benchmarks for low-resource languages, even with human labor, which further strains existing benchmark resources. This growing demand for scalable and dynamic evaluation methods has thus spurred the emergence of automated benchmark creation using LLMs, or "LLM-as-a-benchmark"(Farchi et al., 2024; Maheshwari et al., 2024; Pombal et al., 2025a). These approaches promise a cost-effective and agile alternative to human curation, potentially revolutionizing how models are ranked and progress is measured in natural language processing.

While LLM-as-a-benchmark offers a promising avenue for efficient model ranking, our research uncovers a significant limitation: LLM-generated benchmarks exhibit strong self-bias, particularly with strong frontier models and low-resource XX→En translation. This systemic bias disproportionately favors the LLM that created the benchmark, suggesting that these benchmarks may reflect the generator's inherent biases rather than providing an objective assessment of a model's true capabilities. In this paper, we formally define self-bias based on the statistical concept of estimator bias. Here, the evaluator model's estimated ranking serves as the empirical expected value, while the mean rankings from peer models act as a proxy for the true estimation. Self-bias is then quantified as the difference between the estimated value and the true (proxy) value (Xu et al., 2024).

We decompose self-bias in LLM-as-a-benchmark into two constituents: LLM-as-a-testset and LLM-as-an-evaluator. While LLM-as-an-evaluator has been extensively studied—with biases often attributed to self-recognition (Panickssery et al., 2025) or genuine quality improvements (Chen et al., 2025)—the origins of self-bias in LLM-as-a-benchmark remain underexplored. Our primary contri-

---

[0]Correspondence to: wendax@google.com, swetaagrawal@google.com, vzouhar@ethz.ch, freitag @google.com, dandeutsch@google.com.

bution is a comprehensive analysis of self-bias attribution within the LLM-as-a-benchmark framework, specifically elucidating its mechanisms across different language directions and source text generation methods.

For translation tasks, we investigate these mechanisms and present three key findings:

First, self-bias in the LLM-as-a-benchmark paradigm stems from dual, additive sources: test data generation (LLM-as-a-testset) and the evaluation method (LLM-as-an-evaluator). Their combined interaction amplifies the overall self-bias, posing a notable challenge to fair evaluation.

Second, our investigation shows that the magnitude of self-bias in LLM-as-a-benchmark is heavily influenced by the LLM's generation capability in the specific source language. This leads to a crucial asymmetry in bias. For instance, we observe significantly greater self-bias in into-English (XX→En) translation directions than in out-of-English (En→XX).

Third, we demonstrate that one attribution of self-bias stems from the LLM's limited generation capability in low-resource languages, which leads to the production of homogeneous source texts characterized by repetitive content and stylistic traits. Our findings show that improving the diversity of these generated source texts can mitigate some of the observed self-bias.

## 2 PRELIMINARIES AND SELF-BIAS DEFINITION

**LLM-as-a-testset** We automatically generate a test set using a generator model, $M_{\text{test}}$. For a given instruction $s$, $M_{\text{test}}$ produces a source text $x$ and an optional reference text $y'$. The source text $x$ is then used as a prompt for the target model under evaluation, $M_{\text{target}}$, which generates an output $y$. Finally, an evaluation metric computes a quality score for $y$ by comparing it against source text $x$. We investigate testset generation self-bias: a critical flaw where a generated test set inherently favors the generator model ($M_{\text{test}}$), leading to inflated scores that do not reflect true model capabilities.

**LLM-as-an-evaluator** An evaluator model, $M_{\text{evaluator}}$, assesses the quality of an output $y$ generated by a target model $M_{\text{target}}$ in response to a prompt $x$, assigning it a score based on predefined criteria. In our experimental setup, the prompts $x$ are sourced from human-authored benchmark datasets. We investigate the phenomenon of evaluator self-bias, where an LLM-based evaluator assigns disproportionately high scores to outputs from its own model, regardless of their actual quality compared to outputs from other models.

**LLM-as-a-benchmark** We define the LLM as Benchmark setting as a paradigm where a single language model serves two roles: it generates a source text $x$ to prompt a target model, $M_{\text{target}}$, and subsequently evaluates the quality of $M_{\text{target}}$'s output. This approach effectively combines the "LLM as Testset Generator" and "LLM as Evaluator" functions. This paper investigates the self-bias inherent in the LLM as Benchmark paradigm.

We now operationalize evaluation self-bias as follows:

$$\text{bias}_{M_i} = \underbrace{\theta_{M_i,M_i}}_{\text{self-ranking}} - \underbrace{\frac{1}{|M|-1} \sum_{M_o \neq M_i} \theta_{M_i,M_o}}_{\text{ranking by other models}} \tag{1}$$

where $M$ is a set of models, $M_i$ is a specific model that is being evaluated by itself, and $M_o$ are other models that evaluate $M_i$. The $\theta_{M_i,M_o}$ is an outcome of evaluating model $M_i$ by model $M_o$. This evaluation outcome can be, for example, number of assigned points averaged across multiple samples, or ranking in a task.

The $\theta_{M_i,M_i}$ is the outcome of model $M_i$ evaluating itself and $\frac{1}{|M|-1} \sum_{M_o \neq M_i} \theta_{M_i,M_o}$ is the average evaluation according to other models. We consider the latter the true estimate of the true performance of $M_i$. The difference between the true evaluation and self-evaluation becomes our quantity of interest, $\text{bias}_{M_i}$. A self-bias score will be negative if a model ranks its own output more favorably than other models would rank that same output. A more negative score indicates a stronger bias by the model towards its own generations.

In our study for the task of machine translation, the role of $M_o$ in $\theta_{M_i,M_o}$ can be either that of input text generation, translation quality evaluator, or both.

## 3 EXPERIMENTAL SETUP

To empirically investigate self-bias, we situate our analysis within the machine translation domain, employing an LLM-as-a-benchmark methodology. Our experiments encompass six low to medium resource language directions: Bemba→English, Kurdish→English, Aymara→English, Luo→English, English→Bemba, and English→Aymara. In high-resource languages, performance is often near-optimal, which obscures the subtle disparities required for reliable bias detection. For instance, frontier LLMs frequently rate each other's outputs as nearly perfect. This phenomenon motivates our current investigation.

Our analysis involves three state-of-the-art LLMs at translation: Gemini 2.5 Pro, GPT-4.1, and Claude 3 Opus, which are tested under three experimental conditions:

- LLM-as-a-testset Generator: LLMs generate source texts, and their translation quality is subsequently assessed using the MetricX quality estimation metric (QE), which scores based on both source and translation (Juraska et al., 2024).
- LLM-as-an-evaluator: LLMs function as evaluators, scoring translations of canonical source texts drawn from the FLORES-200 benchmark (Team et al., 2022).
- LLM-as-a-benchmark: In our primary configuration, LLMs perform both source text generation and the subsequent evaluation of translations produced by all models. This LLM-as-a-benchmark evaluation scores based on source and translation, without requiring a reference text. The default setup for source text generation involves generating both source and reference texts, while a truly reference-free variant of this approach is analyzed in Section 5.

To investigate the LLM-as-a-testset and LLM-as-a-benchmark, we generated 200 source texts for each language direction and obtained translations from all three LLMs. For studying LLM-as-an-evaluator independently, we sampled 200 examples per language direction from the FLORES benchmark, subsequently obtaining translations from the same three LLMs. All prompts used for generation, translation, and evaluation are provided in appendix B. Notably, **all evaluations (whether using MetricX QE or LLM-as-an-evaluator) were conducted in a reference-free setting**.

For score normalization and comparability across conditions, we convert raw numerical scores into ranking. Specifically, for each source segment, the outputs from the three generation models are scored by an evaluator. A model's aggregate system score is determined by the mean of its per-segment ranks over the entire dataset.

## 4 DOES THE USE OF LLM-AS-A-BENCHMARK INTRODUCE SELF-BIAS?

This section investigates the self-bias inherent in the LLM-as-a-benchmark paradigm, where models generate a test set and evaluate their own outputs. To generate our dataset, we employed the prompt proposed by Pombal et al. (2025b) in their LLM-as-a-testset framework. This prompt instructs the models to generate source-reference text pairs covering diverse topics and varying in length (See appendix B for more details).

Table 1 quantifies the self-bias for Gemini-2.5-pro, GPT-4.1, and Claude-Opus-4 for the XX→En direction. In our framework, a negative bias score indicates preferential treatment. The consistently negative diagonal scores confirm that **these models systematically favor their own translations** in four different XX-EN language directions, except Gemini-2.5-Pro on Kurdish to English translation.

To understand what contributes to self-bias in LLM-as-a-benchmark, we examine the self-bias of LLM-as-a-test and LLM-as-an-evaluator as distinct components. We conducted an ablation study presented in Table 2 which isolates the impact of LLM-as-a-testset, where models generate test sets for a fixed external metric (MetricX) and LLM-as-a-benchmark, where the model acts as an evaluator on the Flores test set. **Both setups result in measurable self-bias**. The LLM-as-a-benchmark setting shows a larger self-bias than the LLM-as-a-testset, which suggests that using an LLM both as a generator and evaluator can result in a compounding effect.

**Where does the self-bias stem from?** The simultaneous generation of source texts and reference translations, as proposed in Pombal et al. (2025a), could result in two distinct types of biases: a model's inherent stylistic "dialect" in the source language, and a translatability bias where the model generates source sentences it already knows it can translate well.

| Bemba→En. | LLM-as-a-benchmark | | |
|---|---|---|---|
| Translator | **Gemini** | **GPT** | **Claude** |
| **Gemini** | -0.591 | 0.161 | 0.430 |
| **GPT** | -0.145 | -0.202 | 0.347 |
| **Claude** | 0.102 | 0.515 | -0.617 |

| Aymara→En. | LLM-as-a-benchmark | | |
|---|---|---|---|
| Translator | **Gemini** | **GPT** | **Claude** |
| **Gemini** | -0.157 | 0.180 | -0.022 |
| **GPT** | 0.232 | -0.315 | 0.083 |
| **Claude** | 0.253 | 0.365 | -0.617 |

| Luo→En. | LLM-as-a-benchmark | | |
|---|---|---|---|
| Translator | **Gemini** | **GPT** | **Claude** |
| **Gemini** | -0.300 | 0.120 | 0.180 |
| **GPT** | 0.385 | -0.508 | 0.123 |
| **Claude** | 0.188 | 0.352 | -0.540 |

| Kurdish→En. | LLM-as-a-benchmark | | |
|---|---|---|---|
| Translator | **Gemini** | **GPT** | **Claude** |
| **Gemini** | 0.005 | 0.215 | -0.219 |
| **GPT** | 0.173 | -0.150 | -0.023 |
| **Claude** | -0.095 | 0.520 | -0.425 |

Table 1: Bias estimation of using Gemini-2.5-Pro, GPT4.1 and Claude-Opus-4 as LLM-as-a-benchmark on Bemba→English, Aymara→English, Luo→English and Kurdish→English. For each column, each LLM is used as both testset generator and evaluator (LLM-as-a-benchmark). For each each row, LLM is used as generation model. Self-bias estimation is the diagonal line across all the tables. If the bias score is below 0, this indicates that system acting as the LLM-as-a-benchmark prefers its own translation system. Three LLMs display self-bias across four language directions except Gemini-2.5-Pro on Kurdish-to-English translation.

| XX→En | **Gemini** | **GPT** | **Claude** |
|---|---|---|---|
| **LLM-as-a-testset** | -0.124 | -0.239 | -0.093 |
| **LLM-as-a-benchmark** | -0.261 | -0.294 | -0.550 |
| **LLM-as-an-evaluator** | -0.302 | -0.443 | -0.303 |

Table 2: Self-bias for three models in three scenarios: using an LLM to generate the test set ("LLM-as-a-testset"), using an LLM for both testset generation and evaluation ("LLM-as-a-benchmark"), and using an LLM for evaluation ("LLM-as-an-evaluator"). The last row is separate because it operates on different data. Across all scenarios, all models consistently prefer their own outputs.

When generating both src+ref texts, the LLM might generate texts that it knows how to translate, avoiding source texts for which it lacks confident translation mappings. We confirm this translatability bias by assessing translation quality using MetricX-QE for LLM-generated test sets under two scenarios: src-only and src+ref. As Table 3 illustrates, src+ref generation consistently yielded superior translation performance for both Bemba→English and Aymara→English. This strongly supports our hypothesis that **co-generation introduces a significant confounding variable, essentially tailoring the source material to the model's translation strengths.**

The superior performance on 'src+ref' texts thus clearly stems from an artificially easier, pre-filtered source, not from enhanced translation capability. While this translatability bias is noteworthy, our primary interest lies in the more fundamental source effect bias. This is introduced as an LLM potentially might generate texts in its own native "dialect", confirming certain linguistic styles and patterns to which it has preference towards. This text when translated by the same model results in higher quality translations. Furthermore, since LLMs have the ability to recognize their own outputs (Panickssery et al., 2025), they also assess them of higher quality. For higher resource languages like English, many of the current LLMs generate text that are of similar quality and diversity whereas for the lower-resource languages, as the model's distribution is less-developed due to access to finite datasets during training, for these languages, the model's "native dialect" is far narrower and more repetitive.

To isolate this effect, we conduct additional ablations and study self-bias where a) the model is asked to generate just the source text without generating a reference translation and b) we also study bias in higher-resources EN-XX settings. This allows us to systematically study the pure impact of the "dialect" bias, i.e. it shows how much easier it is for the model to translate text that conforms to its own general monolingual patterns, even without the translation task in mind.

| Bemba→En | **Gemini** | **GPT** | **Claude** | | Aymara→En | **Gemini** | **GPT** | **Claude** |
|---|---|---|---|---|---|---|---|---|
| **Source-only** | -8.49 | -5.33 | -4.07 | | **Source-only** | -12.90 | -10.40 | -8.29 |
| **Source+Ref** | -2.73 | -4.76 | -3.89 | | **Source+Ref** | -7.54 | -9.60 | -8.13 |

Table 3: Translation quality improves when source texts are co-generated with their reference translations rather than generated in isolation. This disparity strongly suggests that the paired generation process acts as a confounding variable, creating source texts that are pre-aligned with the model's capabilities. MetricX negative values indicate errors.

| Bemba→En | LLM-as-a-benchmark | | | | Aymara→En | LLM-as-a-benchmark | | |
|---|---|---|---|---|---|---|---|---|
| | **Gemini** | **GPT** | **Claude** | | | **Gemini** | **GPT** | **Claude** |
| **Gemini** | -0.515 | 0.055 | 0.460 | | **Gemini** | -0.293 | 0.285 | 0.008 |
| **GPT** | 0.235 | -0.253 | 0.018 | | **GPT** | 0.308 | -0.330 | 0.022 |
| **Claude** | 0.220 | 0.543 | -0.763 | | **Claude** | 0.015 | 0.473 | -0.488 |

Table 4: This table presents the self-bias estimations for Gemini-2.5-pro, GPT-4.1, and Claude-Opus-4 in the LLM-as-a-benchmark setup for Bemba→English and Aymara→English. Despite the data generation pipeline being limited to producing only source texts, all three LLMs still exhibit self-bias across both language pairs.

## 5 HOW DOES SOURCE TEXT GENERATION IMPACT SELF-BIAS?

In this section, we first establish that source-only generation inherently leads to self-bias (Section 5.1). Our investigation then reveals that this bias is partially driven by models' tendency to produce homogeneous and repetitive content, even when explicitly prompted for diverse topics (Section 5.2). We further demonstrate that limited diversity source text generation is one attribution of this self-bias (Section 5.3). Finally, we show that improving source text diversity can mitigate some of the observed self-bias across all three LLMs (Section 5.4). Due to budget constraints, this section's study focuses on Aymara→English and Bemba→English.

### 5.1 SOURCE TEXT GENERATION LEADS TO SELF-BIAS

As shown in Table 4, a measurable self-bias persists even when the data generation pipeline is constrained to produce only source texts. This self-bias is highly localized to each model's own output, evidenced by negative bias scores appearing exclusively on the diagonal. Source only text generation is not an alternative solution to replace "src+ref" generation.

### 5.2 WHAT MAKES SOURCE TEXT GENERATION EXHIBIT BIAS?

In this section, we investigate how an LLM's source text diversity and stylistic characteristics contribute to self-bias. This bias manifests as a model generating content and style highly consistent with its own inherent "dialect": for a given model, its source texts display non-trivial high similarity to other texts it produces (even under different topics) when compared to texts from other models. To quantify this phenomenon, we define within-model similarity and cross-model similarity.

Let $M = \{M_1, M_2, M_3\}$ be the set of LLMs (Gemini-2.5-pro, GPT4.1, Claude-Opus-4). Each model $M_k \in M$ generates $N$ source texts, $S_k = \{s_{k,1}, \ldots, s_{k,N}\}$, where $j$ in $s_{k,j}$ denotes a topic ID. We quantify text similarity using chrF@K, measuring how well source texts from one model align with those from another (or the same) model. This encompasses both within-model ($M_A = M_B$) and cross-model ($M_A \neq M_B$) comparisons.

For a source text $s_{A,i} \in S_A$, its chrF@K similarity to model $M_B$ is calculated by: 1. Computing pairwise chrF scores, chrF$(s_{A,i}, s_{B,j})$, against all $s_{B,j} \in S_B$. (Self-matches are excluded when $M_A = M_B$.) 2. Selecting the $K = 5$ highest chrF scores, $\{\text{chrF}_1, \ldots, \text{chrF}_K\}$, and averaging them:

$$\text{chrF@K}(s_{A,i}, M_B) = \frac{1}{K} \sum_{k=1}^{K} \text{chrF}_k \qquad (2)$$

| Bemba→En | chrF across prompts + models | | |
| --- | --- | --- | --- |
| | **Gemini** | **GPT** | **Claude** |
| **Gemini** | 35.76 | 32.70 | 32.03 |
| **GPT** | 32.28 | 37.78 | 31.63 |
| **Claude** | 36.42 | 35.99 | 38.64 |

(Data from)

| Aymara→En | chrF across prompts + models | | |
| --- | --- | --- | --- |
| | **Gemini** | **GPT** | **Claude** |
| **Gemini** | 37.78 | 32.19 | 32.58 |
| **GPT** | 33.52 | 40.79 | 32.44 |
| **Claude** | 39.56 | 37.04 | 42.28 |

(Data from)

Table 5: presents average chrF@K similarity scores, differentiating within-model (diagonal entries) from cross-model (off-diagonal entries) comparisons. The results show that each model maintains a significantly higher similarity to its own generated content (even across varied topics) than to texts from other models. This within-model similarity suggests that, rather than achieving true textual diversity as instructed, models tend to repeat content and style from their limited knowledge for low-resource languages like Bemba and Aymara.

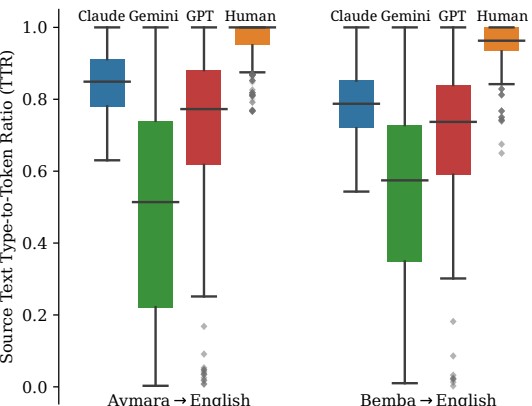

| **Model** | **Degeneration Ratio (%)** |
| --- | --- |
| Gemini | 22.3 |
| GPT4.1 | 5.8 |
| Claude | 0 |

| **Model Pair on TTR** | **Cohen's D** |
| --- | --- |
| Gemini×GPT4.1 | 0.777 |
| Gemini×Claude | 1.487 |
| Claude×GPT4.1 | 0.663 |

Figure 1: The left panel illustrates the Type-Token Ratio (TTR) of source texts (three LLMs vs. FLORES human references), indicating higher lexical diversity in human texts and distinct TTR profiles among LLMs. Bottom right, pairwise Cohen's D values exceeding 0.5 quantify these TTR distributional differences as moderate to substantial. Top right, estimated degeneration ratios across two language directions reveal differential degeneration levels in the three models during source text generation (Degeneration is defined as 4-gram repeating more than 10 times).

Finally, to obtain a single average similarity score for $M_A$'s perspective on $M_B$'s style, we average these chrF@K scores across all $s_{A,i} \in S_A$:

$$\text{Avg. chrF@K}(M_A, M_B) = \frac{1}{N} \sum_{i=1}^{N} \text{chrF@K}(s_{A,i}, M_B) \tag{3}$$

This process yields three final scores for each model $M_A$: one for its average within-model similarity (to itself), and two for its average cross-model similarities to the other two LLMs. Table 5 presents the average chrF@K similarity scores, with diagonal entries representing within-model similarity and off-diagonal entries showing cross-model similarities. The results clearly demonstrate that each model exhibits a significantly higher similarity to its own generated content (even across different topics) compared to texts generated by other models. This strong within-model consistency suggests that, rather than generating truly diverse texts following instructions, the models tend to reproduce content and style from their limited knowledge for low-resource languages like Bemba and Aymara.

### 5.3 UNDERLYING CAUSES: LIMITED DIVERSITY AND QUALITY IN SYNTHETIC LOW-RESOURCE LANGUAGE SOURCE TEXTS

To further support and explain our findings regarding within and cross model similarities (as quantified by chrF@K similarity), we hypothesize that the underlying cause lies in the limited diversity and quality of the generated source texts themselves. To test this, we measure several key linguistic properties of these generated source texts. Specifically, we assess lexical diversity (Type-Token Ratio; TTR) and degeneration ratio, comparing each model's output against each other.

| | Gemini | GPT | Claude | | | Gemini | GPT | Claude |
|---|---|---|---|---|---|---|---|---|
| **English→Bemba** | -0.145 | 0.103 | 0.138 | **English→Aymara** | | 0.016 | -0.092 | 0.069 |
| **Bemba→English** | -0.591 | -0.202 | -0.617 | **Aymara→English** | | -0.157 | -0.315 | -0.617 |

Table 7: Out-of-English directions exhibit less self-bias in LLM-as-a-benchmark. We observe a greater self-bias when LLMs generate texts in out-of-English directions compared to into-English directions. The magnitude of self-bias for English→Aymara and English→Bemba is consistently less than 0.15.

Figure 1 illustrates the differences in TTR and degeneration ratio across the three evaluated models. All three LLMs exhibit distinctive TTR distributions. We quantified these distributional differences using Cohen's D (Cohen, 1988) (see Appendix A for an introduction), with pairwise Cohen's D values exceeding 0.5 (bottom right panel) indicating moderate to substantial divergence. All three LLMs, including Claude, demonstrate less lexical diversity than human-written benchmarks[1]. This supports prior work (Yu et al., 2023) on data diversity's role in mitigating systematic bias (We will discuss more details in the next section).

Further inspection reveals that the notably low diversity in sources from Gemini-2.5-pro and GPT4.1 often stems from degeneration. **We quantified this by counting repeating n-grams, marking texts with ≥10 repeating 4-grams as degenerated**. As shown in Figure 1 (top right panel), both models exhibit varying degeneration levels across language directions. Interestingly, generation-time degeneration patterns are often model-specific; translation quality from such sources can be improved when the translation model is the same as the generation model (Appendix D), suggesting better recognition and compensation for these characteristic flaws.

## 5.4 IMPROVING ON DIVERSITY CAN REDUCE SELF-BIAS

We hypothesize that the one attribution of self-bias is the lack of diversity in the generated source texts. To investigate this, we conducted an ablation study on source texts using our established within-model similarity metric. As defined in Equation (3) ($M_A = M_B$), a high within-model similarity score indicates a model's tendency to repeat its own content and style, even when prompted for diverse topics, thereby generating less diverse texts. For each of the three LLMs, we selected three subsets of 50 source texts from the total 200: those with high within-model similarity (representing low lexical diversity), those with low within-model similarity (representing high lexical diversity), and a randomly selected subset for control.

| | Self-bias Estimation | | |
|---|---|---|---|
| Subset | **Gemini** | **GPT** | **Claude** |
| **Max chrF** | -0.400 | -0.280 | -0.685 |
| **Random** | -0.342 | **-0.256** | -0.616 |
| **Min chrF** | **-0.250** | -0.265 | **-0.600** |

Table 6: Self-bias of LLM-as-a-benchmark for subsets of LLM-generated source texts: those with the highest within-model chrF similarity (lowest lexical diversity), those with the lowest within-model chrF similarity (highest lexical diversity), and a randomly selected baseline. Diverse source texts mitigate self-bias. Self-bias is averaged for Aymara to English and Bemba to English.

Table 6 demonstrates that source texts corresponding to high within-model similarity (Max chrF) consistently exhibit the highest self-bias across all three LLMs, surpassing the bias observed in the random 50-sample baseline. Conversely, source texts with low within-model similarity (Min chrF) consistently lead to reduced or comparable self-bias compared to random and the Max chrF subset. This compellingly suggests that generating more diverse source texts can mitigate self-bias.

## 6 IMPACT OF TRANSLATION DIRECTION ON SELF-BIAS

In Table 7, we show that the out-of-English directions exhibit lower self-bias in LLM-as-a-benchmark. We observe a lower self-bias when LLMs generate texts in out-of-English directions compared to into-English directions. The magnitude of self-bias for English→Aymara and English→Bemba is consistently less than 0.15.

---

[1]We selected 200 source texts from Flores-200 Aymara and Bemba texts.

|  | LLM-as-a-testset | | | |  | LLM-as-an-evaluator | | |
|---|---|---|---|---|---|---|---|---|
|  | **Gemini** | **GPT** | **Claude** |  |  | **Gemini** | **GPT** | **Claude** |
| **English→XX** | 0.024 | 0.054 | -0.076 |  | **English→XX** | -0.110 | 0.049 | 0.099 |
| **XX→English** | -0.174 | -0.239 | -0.093 |  | **XX→English** | -0.302 | -0.443 | -0.303 |

Table 8: Self-bias decomposition in LLM-as-a-benchmark. Decomposing LLM-as-a-benchmark into LLM-as-a-testset and LLM-as-an-evaluator reveals that both components exhibit greater self-bias for into-English directions compared to out-of-English directions. This indicates that for XX→English, LLMs tend to generate self-favorable source texts and assign higher scores to their own outputs. Conversely, self-bias is significantly less pronounced in En→XX directions.

|  | Source-Only | | | Source+Ref | | |
|---|---|---|---|---|---|---|
|  | **Gem.×Claude** | **Gem.×GPT** | **Claude×GPT** | **Gem.×Claude** | **Gem.×GPT** | **Claude×GPT** |
| English→XX | 0.076 | 0.190 | 0.111 | 0.164 | **0.259** | 0.081 |
| XX→English | **1.487** | **0.777** | **0.663** | **0.597** | 0.111 | **0.556** |

Table 9: Source Text Type-to-Token Ratio distribution differences between models. We examined the Type-to-Token Ratio (TTR) distributions of source texts generated by different models. We found that for texts generated in English (as source for En→XX translation), all models exhibit relatively similar TTR distributions. This similarity is less pronounced for source texts generated in other languages (for XX→En translation). This suggests that the models generate English source texts with more consistent lexical diversity compared to other languages.

Table 8 presents a decomposition of LLM-as-a-benchmark into its LLM-as-a-testset and LLM-as-an-evaluator components, elucidating the sources of self-bias. Both components consistently exhibit a more pronounced self-bias in into-English (XX→En) directions than in out-of-English (En→XX) directions. This observation highly suggests that in XX→En generation, the LLM-as-a-testset produces source texts containing intrinsic linguistic features that offer an advantage to its own translation system. Simultaneously, the LLM-as-an-evaluator appears more sensitive to these self-generated patterns when judging its own XX→En outputs, leading to systematically higher scores for these directions compared to En→XX.

**Why does translation asymmetry exist for self-bias?** To answer the translation asymmetry in self-bias, we leverage the findings that we had in previous section and examine the chrF similarity and type-to-token ratio distributions for English source texts. In Table 9, we showed that TTR in English as source texts are more similar or consistently generated across all three LLMs. However, the similarity is less pronounced for source texts generated in XX languages. This suggests that the models generate English source texts with more consistent lexical diversity compared to other languages, which could be the attribution for English→XX direciton has less self-bias. Table 11 demonstrates that neither Gemini-2.5-pro nor Claude-Opus-4 exhibit biased source text similarity (represented by diagonal entries), a contrast to XX→En translations. Specifically, Claude-generated outputs show cross-model chrF similarities with GPT4.1 and Gemini-2.5-pro that are comparable to its within-model similarity. It's important to note, however, that direct chrF differences are not strictly comparable between English and XX texts due to inherent linguistic variations. Appendix Appendix C demonstrates that the TTR distribution of English source texts is closer to human-written texts compared to that of low-resource (XX) language source texts.

## 7 MERITS OF LLM-AS-A-BENCHMARK

**LLM-as-a-benchmark can still benefit open source models** Table 10 reveals distinct evaluation patterns for frontier LLMs (Gemini-2.5-Pro, GPT-4.1, Claude-Opus-4). We observe that these models consistently rank open-source models (Gemma3-27B, Mistral-large-2411, Qwen3-32B) with low intrinsic bias, aside from GPT-4.1's specific bias toward Qwen3-32B. This consistency, which aligns with findings in LLM-as-a-benchmark evaluation (Pombal et al., 2025a), supports the continued utility of LLM-as-a-benchmark for fostering rapid iteration in open-source model development. However, a significant concern arises from the substantial self-bias exhibited by these frontier models

| Translator | Bemba | LLM-as-a-benchmark (Rank) | | | Bemba | LLM-as-a-benchmark (Bias) | | |
|---|---|---|---|---|---|---|---|---|
| | | Gemini | GPT | Claude | | Gemini | GPT | Claude |
| | **Gemma3** | 1.125 | 1.110 | 1.160 | **Gemma3** | -0.01 | 0.04 | -0.03 |
| | **Mistral** | 1.675 | 1.623 | 1.630 | **Mistral** | 0.05 | -0.02 | -0.03 |
| | **Qwen3** | 2.335 | 2.558 | 2.285 | **Qwen3** | -0.09 | -0.16 | 0.25 |

Table 10: On the left, we observe consistent rankings of Gemma3-27B, Mistral-large-2411, and Qwen3-32B by Gemini-2.5-Pro, GPT-4.1, and Claude-Opus-4. The right table illustrates minimal bias from frontier models towards these open-source models, except for GPT-4.1's bias towards Qwen3-32B. This consistency in ranking open-source models aligns with (Pombal et al., 2025a). However, significant self-bias is evident when frontier models rank each other.

when ranking their peers. This self-bias indicates that automated benchmarking approaches, while may be effective for open-source models, may yield skewed and unreliable evaluations for frontier models, necessitating careful consideration in their application to advanced model development.

## 8 RELATED WORK

**Automatic Benchmark Creation** As existing benchmarks become increasingly saturated by the rapid advancements in LLM capabilities Glazer et al. (2025), the field has shifted towards exploring automatic benchmark construction using LLMs. This approach generally involves an LLM generating benchmark data from task instructions (LLM-as-a-benchmark) Pombal et al. (2025b), subsequently used for ranking various LLM models. The efficacy of such automatically generated benchmarks is typically evaluated either by assessing benchmark agreement (Perlitz et al., 2024) or by comparing their ranking correlations with human-written benchmarks (Pombal et al., 2025b). Depending on the specific task requirements, this automated creation process can encompass diverse methods, such as automating software environment setups for repositories (Vergopoulos et al., 2025), constructing new user prompts from existing data (Li et al., 2024), or synthesizing test sets through complex prompt workflows (Sprague et al., 2024; Zouhar et al., 2025). Concurrently, the evaluation of model outputs varies based on task verifiability, utilizing simple accuracy metrics for objective tasks (Sprague et al., 2024) or employing LLMs as judges for more nuanced prompt-answer pair evaluations (Xu et al., 2023; Pombal et al., 2025b).

**Self-bias in LLM** The "LLM-as-a-benchmark " paradigm is susceptible to self-bias from two sources: the LLM acting as an evaluator and the LLM generating the testset. While most prior work has focused on the evaluator, a well-documented issue is the tendency of an LLM judge to systematically favor its own outputs (Xu et al., 2024; Panickssery et al., 2025). This preference is often linked to the judge's familiarity with its own stylistic patterns or a bias towards low-perplexity text (Wataoka et al., 2025). Although this behavior could sometimes reflect genuine quality improvements (Chen et al., 2025), a judge's reliability is questionable for problems it cannot solve itself (Krumdick et al., 2025). Our work extends this analysis by investigating the overlooked self-bias from testset generation and, crucially, the additive effects when both biases are present. While Yuan et al. (2025) also address biases in automated benchmarks, their mitigation strategies are limited to verifiable tasks like math reasoning are not immediately applicable to generative tasks where evaluating model success is a task in itself.

## 9 CONCLUSION

Our work formally defines and quantifies self-bias in LLM-as-a-benchmark, attributing its origin to a synergistic interplay between LLM-as-a-testset and LLM-as-an-evaluator. We show this bias is influenced by the LLM's source language proficiency, appearing more strongly in into-English translation. Moreover, we observe that low diversity in source text is an attribution to self-bias. Our results suggest that improving the diversity of these generated source texts can mitigate some of the observed self-bias. Despite these challenges, we identify potential use cases where LLM-as-a-benchmark remains valuable. It reliably ranks less competitive models, exhibiting smaller bias and consistent rankings in such scenarios. Moreover, for languages where LLMs generate high-quality source texts (e.g., out-of-English translations), LLM-as-a-benchmark presents less risk in self-bias.

## 10 Reproducibility Statement

We access Gemini-2.5-Pro, Claude-Opus-4@-20250514, and GPT4.1@2025-04-14 via publicly available commercial APIs. The FLORES benchmark is publicly available. Gemma3-27B, Mistral-large-2411, and Qwen3-32B are publicly available on Huggingface. We plan to release code and data upon publication to facilitate further research. All results presented in this paper are reproducible. Gemini-2.5-Pro was used to polish the writing of this paper.

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

# A  COHEN'S D: EFFECT SIZE FOR MEAN DIFFERENCES

Cohen's d (Cohen, 1988) is a widely used standardized effect size measure to quantify the difference between two means, expressed in standard deviation units. It is particularly useful when comparing the central tendency of two groups (e.g., two sets of scores, two distributions) and interpreting the practical significance of their difference, independent of sample size.

Given two groups, Group 1 and Group 2, with means $\bar{x}_1$ and $\bar{x}_2$ and standard deviations $s_1$ and $s_2$ respectively, Cohen's d is typically calculated as:

$$d = \frac{\bar{x}_1 - \bar{x}_2}{s_p}$$

where $s_p$ is the pooled standard deviation, calculated as:

$$s_p = \sqrt{\frac{(n_1 - 1)s_1^2 + (n_2 - 1)s_2^2}{n_1 + n_2 - 2}}$$

$n_1$ and $n_2$ are the sample sizes for Group 1 and Group 2.

Interpretation Guidelines (Cohen's conventions):

- $d = 0.2$: Small effect
- $d = 0.5$: Medium effect
- $d = 0.8$: Large effect

# B  PROMPTS USED FOR LLM-AS-A-BENCHMARK

All prompts for both LLM-as-a-testset and LLM-as-an-evaluator are included. To ensure fair comparisons, these prompts are adapted from the zero-shot benchmark paper by (Pombal et al., 2025a). The LLM-as-a-testset prompts incorporate a few randomly chosen seed variables—such as length, topic, subtopic, and style—to guarantee diverse generated source texts. A complete list of options for each seed variable is provided below.

**Prompt for LLM-as-a-Testset (source + reference generation) for translation.**   You are a
multilingual content creator and translation expert.  Your task is to generate a
comprehensive translation exercise based on the given attributes.  Follow these
instructions carefully:
1.  Review the following input variables:
- Source language:  SOURCE LANGUAGE
- Target language:  TARGET LANGUAGE
- Topic:  topic
- Subtopic:  subtopic
- Source length:  length
- Style:  style
2.  Generate a source text:  Create an original text in the source language,
adhering to the specified topic, subtopic, and length.  The text should be
coherent, informative, and suitable for translation.
3.  Generate a reference translation:  Produce a high-quality, fluent translation
of the source text in the target language.
This translation should serve as a reference for evaluating other translations.
IT IS CRUCIAL THAT THE REFERENCE TRANSLATION SOUNDS NATURAL IN THE TARGET
LANGUAGE. Format your output as follows:
<START OF SOURCE>
INSERT THE SOURCE TEXT HERE
<END OF SOURCE>
<START OF REFERENCE TRANSLATION>

```
648   INSERT THE REFERENCE TRANSLATION HERE
649   <END OF REFERENCE TRANSLATION>
650   Ensure that your response is comprehensive, coherent, and follows all the
651   instructions provided above. Abide strictly by the requested format and
652   generated until the end of the requested output. Only generate source and
653   reference translation. Do not generate any other text such as reasoning or
654   explanations.
655   <START OF SOURCE>
656
657
658   Prompt for LLM-as-a-Testset (source only) for translation.   You are a multilingual
659   content creator and translation expert. Your task is to generate a comprehensive
660   translation exercise based on the given attributes. Follow these instructions
      carefully:
661   1. Review the following input variables:
662   – Source language: SOURCE LANGUAGE
663   – Topic: topic
664   – Subtopic: subtopic
665   – Source length: length
666   – Style: style
667   2. Generate a source text: Create an original text in the source language,
668   adhering to the specified topic, subtopic, and length. The text should be
669   coherent, informative, and suitable for translation.
670   Format your output as follows:
671   <START OF SOURCE>
672   INSERT THE SOURCE TEXT HERE
673   <END OF SOURCE>
674   Ensure that your response is comprehensive, coherent, and follows all the
      instructions provided above.
675   Abide strictly by the requested format and generated until the end of the
676   requested output. Only generate source text. Do not generate any other text
677   such as reasoning or explanations.
678   <START OF SOURCE>
679
680   Topics.      "Tech Innovation", "Global Markets", "Environmental Policy", "Public
681   Health", "Urban Development", "International Relations", "Education Reform",
682   "Cultural Trends", "Scientific Discoveries", "Economic Policy", "Sports
      Industry", "Media & Entertainment", "Workplace Transformation", "Transportation
683   & Mobility", "Food & Agriculture", "Medical & Healthcare", "Legal & Compliance",
684   "E-commerce & Retail", "Financial Services", "Gaming & Software", "Marketing
685   & Advertising", "Government Documentation", "Academic Research", "Patents &
686   Intellectual Property", "Manufacturing & Safety", "Tourism & Hospitality",
687   "Religious & Cultural Studies", "Insurance & Risk Management", "Consumer
688   Electronics", "Pharmaceutical Industry", "Fashion & Apparel", "Beauty &
689   Cosmetics", "Home & Living", "Automotive Industry", "Social Media", "Dating
690   & Relationships", "Parenting & Family", "Arts & Culture", "Music Industry",
691   "Film & Cinema", "Books & Literature", "Food & Cuisine", "Sports & Recreation",
692   "Fitness & Wellness", "Mental Health", "Architecture & Design", "Real Estate",
693   "Telecommunications", "Renewable Energy", "Space Exploration", "Wildlife &
      Nature", "Weather & Climate", "History & Heritage", "Politics & Governance",
694   "NGOs & Nonprofits", "New York City", "London", "Tokyo", "Paris", "Berlin",
695   "Singapore", "Dubai", "São Paulo", "Sydney", "Mumbai", "Madrid", "Lisbon",
696   "Stockholm", "Amsterdam", "Seoul", "Japan", "France", "Germany", "Brazil",
697   "India", "Italy", "Spain", "China", "United Kingdom", "Portugal", "Poetry"
698
699   Subtopics.         "Poetry": [ "Modernism", "Contemporary", "Modernism", "Haiku",
700   "European Poetry", "Asian Poetry", "Theme identification", ], "Tech Innovation":
701   [ "Artificial Intelligence", "Quantum Computing", "Robotics", "5G/6G Networks",
      "Biotechnology", "Green Tech", "Edge Computing", "Cybersecurity", ], "Global
```

Markets": [ "Stock Exchanges", "Cryptocurrency", "International Trade", "Foreign
Investment", "Commodity Markets", "Emerging Markets", "Foreign Exchange", "Market
Regulations", ], "Environmental Policy": [ "Carbon Trading", "Renewable Energy
Initiatives", "Wildlife Protection", "Urban Planning", "Waste Management",
"Climate Agreements", "Marine Conservation", ], "Public Health": [ "Disease
Prevention", "Healthcare Systems", "Vaccination Programs", "Mental Health
Services", "Maternal Health", "Epidemiology", "Health Technology", ], ......

**Styles.** "creative", "concise", "technical", "formal", "informal", "narrative",
"persuasive", "descriptive", "analytical", "humorous", "poetic", "casual",
"academic", "journalistic", "neutral", "elaborate", "minimalist", "rushed"

**Lengths.** "short", "medium"

**Prompt for LLM-as-an-Evaluator for translation.** You are an expert judge evaluating
translation quality. You will be presented with: – An original text – A
translation to evaluate

Rate the translation on a scale of 1–6 based on these key criteria: – Accuracy
and fidelity to source – Grammar and language correctness – Natural flow and
readability – Terminology consistency – Completeness of translation – Technical
precision

Scoring Rubric:

6 – Outstanding – Perfect accuracy with source meaning – Flawless grammar and
language use – Reads completely naturally in target language – Consistent
and precise terminology – Complete translation with no omissions – Excellent
technical accuracy

5 – Excellent – Very accurate rendering of source – Strong grammar with minimal
issues – Natural-sounding translation – Good terminology consistency – Nearly
complete coverage – Strong technical accuracy

4 – Good – Generally accurate translation – Mostly correct grammar – Readable
with some awkward passages – Generally consistent terminology – Minor omissions
only – Adequate technical accuracy

3 – Fair – Some accuracy issues – Notable grammar problems – Often unnatural
phrasing – Inconsistent terminology – Several omissions – Technical inaccuracies
present

2 – Poor – Significant accuracy issues – Frequent grammar errors – Unnatural
throughout – Poor terminology consistency – Major omissions – Many technical
errors

1 – Inadequate – Fails to convey source meaning – Severe grammar issues –
Incomprehensible in target language – No terminology consistency – Incomplete
translation – Technical meaning lost

Format your output as follows: Put detailed explanation between <START OF
FEEDBACK> and </END OF FEEDBACK> Put result between <START OF RESULT> and </END
OF RESULT> Don't provide any other text

<START OF FEEDBACK> Put detailed explanation of the score based on the criteria
here </END OF FEEDBACK>

<START OF RESULT> Put only a number from 1 to 6 here </END OF RESULT> <START
OF SOURCE TEXT> prompt </END OF SOURCE TEXT>

<START OF TRANSLATION> answer </END OF TRANSLATION>

**Prompt for Translation.** You are a professional translator. You are given a
source text in SOURCE LANGUAGE. You need to translate the source text to TARGET
LANGUAGE. Don't include any other text except the translation. Please output
the translation between <START OF TRANSLATION> and <END OF TRANSLATION>. Source
text: SOURCE TEXT

**Prompt for LLM-as-a-Testset for Chatbot Arena.** You are tasked with creating a
diverse and engaging prompt for a chatbot arena. This prompt will be used to
test and compare the capabilities of different language models. Your goal is to

generate a question or prompt that will challenge these models and showcase their strengths or weaknesses.

Here are the input variables you will use to craft your prompt:
– Language: language
– Topic: topic
– Subtopic: subtopic
– Difficulty: difficulty
– Style: style
– Writer: writer
– Writing proficiency: writing_proficiency
– Prompt length: length

Guidelines for creating the prompt:

1. Abide strictly by the input variables provided.

2. Ensure that your prompt is open-ended enough to allow for varied and interesting responses from different language models.

3. Avoid prompts that are overly specific to a particular AI model's capabilities or training data.

4. Create a prompt that is engaging and thought-provoking, encouraging creative or analytical thinking.

Generate output in the following format:

<START OF PROMPT> [Your generated prompt here. IMPORTANT: include only the prompt.] <END OF PROMPT>

Ensure the generated prompt is in the requested language. Remember to abide strictly by the provided input variables and the requested format.

**An example question for Chatbot Arena (English).** In Albion's ledger, where fortunes entwine, what unseen current shapes the City's final line?

**An example question for Chatbot Arena (Bemba).** Finshi fingacitika mu fashion design mu Zambia mu myaka ikumi iileisa? Ndiye, muli busy sana, lelo ndefwaya ukwishiba ifyo mwingalanda pali ifi.

**Prompt for LLM-as-an-Evaluator for Chatbot Arena.** You are an expert judge evaluating response quality for prompts on a variety of topics. You will be presented with:

– An original prompt
– A response to evaluate

Rate the response on a scale of 1–6 based on these key criteria:

– Accuracy and factual correctness
– Comprehensiveness of the answer
– Clarity and organization
– Relevance to the prompt
– Supporting evidence/examples (if applicable)
– Objectivity and balance (if applicable)

Scoring Rubric:
6 – Outstanding

– Exceptionally accurate and well-researched information
– Comprehensive coverage with valuable additional context
– Perfectly structured and crystal clear explanation
– Directly addresses all aspects of the prompt
– Strong supporting evidence and examples
– Maintains perfect objectivity where needed

5 – Excellent

– Highly accurate information
– Thorough coverage of the topic
– Very clear and well-organized
– Closely aligned with the prompt
– Good supporting evidence
– Strong objectivity

```
4 - Good

- Generally accurate information
- Adequate coverage of main points
- Clear organization with minor issues
- Mostly relevant to the prompt
- Basic supporting evidence
- Reasonable objectivity

3 - Fair

- Some inaccuracies present
- Incomplete coverage
- Organization needs improvement
- Partially misses prompt intent
- Limited supporting evidence
- Some bias present

2 - Poor

- Multiple inaccuracies
- Superficial coverage
- Unclear organization
- Largely misses prompt intent
- Minimal supporting evidence
- Clear bias present

1 - Inadequate

- Fundamentally incorrect information
- Severely incomplete
- Disorganized or incomprehensible
- Fails to address prompt
- No supporting evidence
- Extremely biased or misleading

Format your output as follows:
Put detailed explanation between <START OF FEEDBACK> and </END OF FEEDBACK>
Put result between <START OF RESULT> and </END OF RESULT>
Don't provide any other text

<START OF FEEDBACK>
Put detailed explanation of the score based on the criteria here
</END OF FEEDBACK>
<START OF RESULT>
Put only a number from 1 to 6 here
</END OF RESULT>

<START OF PROMPT>
prompt
<END OF PROMPT>

<START OF ANSWER>
answer
<END OF ANSWER>
```

## C WHY DOES TRANSLATION ASYMMETRY EXIST FOR SELF-BIAS?

In Table 12, we showed that model-generated English source texts have TTR distributions more similar to human-written English than model-generated texts in other languages (for XX→En translation) do to their human-written counterparts. This indicates that the lexical diversity of model-generated English source text is closer to that of human-written English source text.

| En→Bemba | chrF across prompts + models | | | En→Aymara | chrF across prompts + models | | |
|---|---|---|---|---|---|---|---|
| | **Gemini** | **GPT** | **Claude** | | **Gemini** | **GPT** | **Claude** |
| **Gemini** | 36.07 | 35.77 | 34.13 | **Gemini** | 36.15 | 35.89 | 34.07 |
| **GPT** | 32.98 | 34.86 | 31.50 | **GPT** | 32.83 | 35.28 | 31.27 |
| **Claude** | 36.75 | 36.47 | 36.85 | **Claude** | 36.83 | 36.62 | 37.14 |

Table 11: Average chrF@K similarity scores, differentiating within-model (diagonal) from cross-model (off-diagonal) comparisons. Unlike XX→En translation, a clear diagonal trend is not observed. For instance, Claude-generated outputs exhibit cross-model chrF similarities with GPT4.1 and Gemini-2.5-pro that are comparable to its within model similarity.

| | Source Text Type-to-Token Ratio distribution differences between model and human | | | | | |
|---|---|---|---|---|---|---|
| | Src Only | | | Src + Ref | | |
| Lang dir | **Gem&Hu** | **GPT&Hu** | **Cla&Hu** | **Gem&Hu** | **GPT&Hu** | **Cla&Hu** |
| En→XX | 1.795 | 1.688 | 1.590 | 1.351 | 1.154 | 1.131 |
| XX→En | **3.187** | **2.335** | **2.232** | **2.306** | **1.826** | **2.010** |

Table 12: We compared the Type-to-Token Ratio (TTR) distributions of model-generated source texts with those of human-generated source texts. We observed that model-generated English source texts (for En→XX translation) have TTR distributions more similar to human-written English than model-generated texts in other languages (for XX→En translation) do to their human-written counterparts. This indicates that the lexical diversity of model-generated English is closer to that of human-written English.

## D    SELF-REPAIR DEGENERATION IN TRANSLATION

In Table 13, we examine Gemini-2.5-pro's self-repair ability when translating its own degenerated source texts. Gemini-2.5-pro more effectively corrects degenerated content it produces during the translation process compared to other models, across both language directions. The observed non-deterministic behavior of GPT-4.1 is likely due to the significantly smaller sample of degenerated source sentences it processed (n=23/400) compared to Gemini-2.5-pro (n=89/400). An example of this self-repair during translation is provided below.

| | Degeneration ratio after translating on degenerated source texts | | | |
|---|---|---|---|---|
| | **Aymara→English** | | **Bemba→English** | |
| Translator | Gemini-2.5-Pro | GPT4.1 | Gemini-2.5-Pro | GPT4.1 |
| Gemini-2.5-Pro | **90.4** | 86.7 | **81.1** | 75.0 |
| GPT4.1 | 94.2 | 93.3 | 86.5 | **62.5** |
| Claude-Opus-4 | 94.2 | **73.3** | 94.6 | 87.5 |

Table 13: Model Self-Repair Ability during Translation of Self-Generated Degenerated Source Texts. This table presents the percentage of translations that retain degenerated content. Gemini-2.5-pro demonstrates a consistently superior ability to correct degenerated content in its self-generated source texts during the translation process, outperforming other models across both language directions. GPT-4.1's observed non-deterministic behavior is potentially attributable to the significantly smaller sample size of degenerated source sentences it processed (n=23/400), compared to Gemini-2.5-pro (n=89/400).

**Example of Gemini's self repair during translation.**

```
Gemini's degenerate source text: Jichhürunakanxa, celularanakax wali
wakiskiripuniwa.  Aka tecnología ukax janiw mayni jaqimpi aruskipt'añatakikiti,
jan ukasti yatiñanak jikxatañataki, anatañataki, ukat yaqha lurawinak lurawinak
lurawinak lurawinak lurawinak lurawinak lurawinak lurawinak lurawinak lurawinak
lurawinak lurawinak lurawinak lurawinak lurawinak lurawinak lurawinak lurawinak
lurawinak lurawinak lurawinak lurawinak lurawinak lurawinak lurawinak lurawinak
[repeated 50 times] lurawinak lurawinak lurawinak lurawinak lurawinak lurawinak
lurawinak lurawinak lurawinak lurawinak lurawinak lurawinak lurawinak lurawinak
lurawinak lurawinak lurawinak lurawinak lurawinak lurawinak lurawinak lurawinak
```

```
lurawinak lurawinak lurawinak lurawinak lurawinak lurawinak lurawinak lurawinak
lurawinak lurawinak lurawinak lurawinak lurawinak lurawinak lurawinak lurawinak
lurawinak lurawinak lurawinak lurawinak lurawinak lurawinak lurawinak lurawinak
lurawinak lurawinak lurawinak lurawinak lurawinak lurawinak lurawinak lurawinak
lurawinak lurawinak lurawinak lurawinak luraw
```

**Gemini's translation:** Nowadays, cell phones are truly necessary. This technology is not only for communicating with other people, but also for finding information, for entertainment, and for carrying out various other activities.

# E  EXTENSION TO NON-TRANSLATION TASKS

We further extend our findings by examining non-translation tasks. Adopting the prompts of (Pombal et al., 2025a), we employ an LLM-as-a-benchmark framework to construct the Chatbot Arena task (Chiang et al., 2024). We detail the prompts for LLM-as-a-testset, LLM-as-an-evaluator, and LLM-as-a-benchmark in appendix B; this diverse prompt set has demonstrated a strong correlation with human-generated benchmarks (Pombal et al., 2025a).

Consistent with our previous experimental setup, we benchmark two distinct settings: high-resource language prompts (English and Korean) versus low-resource language prompts (Bemba and Aymara). As shown in table 14, the results align closely with our findings in the translation task. Gemini-2.5-pro and Claude-Opus-4 exhibit minimal self-bias in English and Korean prompts but show a marked increase in self-bias when prompted in Bemba and Aymara. Specifically, the diagonal entries for Bemba and Aymara are consistently below $0$, indicating significant self-bias.

For high-resource languages (averaged), Gemini, GPT-4.1, and Claude show self-bias scores of $0.025$, $-0.207$, and $-0.017$, respectively. In contrast, for low-resource languages, these scores shift to $-0.301$, $-0.316$, and $-0.125$. This trajectory highlights a sharp increase in self-bias when transitioning from high- to low-resource languages.

| English (Generator) | LLM-as-a-benchmark | | |
|---|---|---|---|
| | **Gemini** | **GPT** | **Claude** |
| **Gemini** | -0.018 | -0.122 | 0.140 |
| **GPT** | 0.035 | -0.205 | 0.170 |
| **Claude** | 0.103 | -0.107 | 0.005 |

| Korean (Generator) | LLM-as-a-benchmark | | |
|---|---|---|---|
| | **Gemini** | **GPT** | **Claude** |
| **Gemini** | 0.068 | -0.060 | -0.007 |
| **GPT** | 0.107 | -0.208 | 0.100 |
| **Claude** | 0.105 | -0.068 | -0.038 |

| Bemba (Generator) | LLM-as-a-benchmark | | |
|---|---|---|---|
| | **Gemini** | **GPT** | **Claude** |
| **Gemini** | -0.385 | -0.010 | 0.396 |
| **GPT** | 0.052 | -0.218 | 0.167 |
| **Claude** | 0.334 | -0.154 | -0.180 |

| Aymara (Generator) | LLM-as-a-benchmark | | |
|---|---|---|---|
| | **Gemini** | **GPT** | **Claude** |
| **Gemini** | -0.216 | 0.346 | -0.130 |
| **GPT** | 0.247 | -0.413 | 0.165 |
| **Claude** | 0.177 | -0.108 | -0.069 |

Table 14: We evaluate LLM-as-a-benchmark on the Chatbot Arena task across high-resource (English and Korean) and low-resource (Bemba and Aymara) settings. The results align closely with our findings in the translation task: all three models exhibit a marked increase in self-bias when prompted in low-resource languages.

