# OpenReview forum: "Deconstructing Self-Bias in LLM-generated translation benchmark"
_ICLR.cc/2026/Conference — Submitted to ICLR 2026_

### Official Review · Reviewer_j9bj · 2025-10-28

**Soundness:** 4
**Presentation:** 4
**Contribution:** 4
**Rating:** 10
**Confidence:** 4

**Summary:**

The paper studies the self-bias in LLM generated benchmarks. LLM generated benchmarks favor the LLM that generated the benchmark dataset. The paper finds strong self-bias in low resource language to English translation. This self-bias comes from two sources, LLM's generation capability in the source language and using LLM as an evaluator. The authors then demonstrate that this self-bias can be attributed to LLM's limited generation capability in the source language.

**Strengths:**

1. The experimental design is sound. It cleanly separates out the two components of self-bias in LLM-as-a-benchmark: benchmark generation and evaluation.
2. The authors perform experiments to determine whether LLM self-bias can be explained by the LLM's preference to generate source texts that are easily translatable by the LLM. Their results of source-only vs source+reference generation conclusively show that this hypothesis is true.
3. The authors also show that even for source-only generation the LLM has low diversity, which leads to the self-bias persisting even in this scenario.
4. Finally the authors also conduct experiment for out of English direction showing lower self-bias. Thus the experiments are comprehensive.

**Weaknesses:**

1. The self bias measurement is based on ranking. This measure might be sensitive to the number of LLMs under consideration (just three in this paper). A larger set of LLMs (including open source multilinguality focused LLMs) may strengthen the results.

**Questions:**

Why were the four low-resource languages chosen? Why not other languages?

---

> ### Author Response · Authors · 2025-11-25
> **For W1**
>
> Thank you for your constructive comments!
>
> We fully agree with the reviewer's suggestion that more LLMs in general could strengthen the results of self-bias estimations. However, our model selection was strictly determined by the requirements of the low-resource Translation task. While we considered available open-source models, such as Gemma3-27B, Mistral-large-2411, and Qwen3, which officially support the target languages, our preliminary analysis and prior studies revealed a significant performance gap. These models are currently not suited for the LLM-as-a-benchmark pipeline in this setting, exhibiting a high degeneration rate when acting as an LLM-as-a-testset (found in source texts). Furthermore, they have a proven evaluation gap compared to the closed-source cluster (Gemini-2.5-pro, GPT4.1, and Claude-Opus-4) when acting as an LLM-as-an-evaluator, as demonstrated by prior work (https://www2.statmt.org/wmt25/pdf/2025.wmt-1.23.pdf).
>
> Since our bias formulation leverages the average estimation of the other two models as the proxy for true performance, selecting models with roughly the same evaluation capability was a scientific necessity to ensure the validity of our results. We leave further extension to other tasks to future work where open-source models have parity with closed-source models in both the LLM-as-a-test and LLM-as-an-evaluator roles to further enhance this study. We will include this discussion in the paper to further clarify our mode choices.

---

> ### Author Response · Authors · 2025-11-25
> **For Q1**
>
> We considered high-resource languages for the analysis as well. However, the focus of our study is on medium/low-resource machine translation, primarily due to the nearly-perfect performance of frontier models in high-resource settings. As shown in the attached table for Chinese-to-English translation, the scoring from Gemini, Claude, and GPT4.1 for all translations is nearly perfect. While models still displayed self-bias in their own generation, its magnitude is quantitatively much smaller. We argue that the near-perfect translation quality of all models makes studying self-bias on high-resource languages less informative for exploring diverse error patterns and evaluating bias towards their own generation. Consequently, we prioritize medium/low-resource language directions, where models can exhibit different error patterns and evaluate them differently due to self-bias. More importantly, the LLM-as-a-benchmark setup is a more practical approach for low-resource translation, as these tasks inherently lack the data to construct a high-quality benchmark.
>
> The table is for Chinese-to-English translation.
>
> | LLM-as-a-Translator \ LLM-as-a-Benchmark | Gemini | Claude | GPT4.1 |
> | :--- | :---: | :---: | :---: |
> | **Gemini** | 6.000 | 5.584 | 5.980 |
> | **Claude** | 5.998 | 5.633 | 5.986 |
> | **GPT4.1** | 5.998 | 5.542 | 5.987 |

---

### Official Review · Reviewer_cz1b · 2025-10-30

**Soundness:** 2
**Presentation:** 3
**Contribution:** 3
**Rating:** 6
**Confidence:** 3

**Summary:**

This submission addresses the self-bias issue in the LLM-as-a-benchmark evaluation, where an LLM can overrate itself when using it as the evaluator or towards the benchmarks that it creates. This submission formally defines and decomposes self-bias into two components: bias from testset generation and bias from evaluation. Focusing on low-resource machine translation, this submission conducts experiments with Gemini-2.5-Pro, GPT-4.1, and Claude-Opus-4 across six translation directions. The results show that self-bias exists in both components and is strongest when combined. This submission further analyses the biases under different aspects, presenting a study that covers multiple points toward the self-bias in LLMs.

**Strengths:**

1. This submission addresses an important issue in LLM evaluation and is generally well written.
2. It provides a formal definition and decomposition of self-bias into testset and evaluator components, which improves the clarity over prior discussions.
3. The experiments cover three major LLMs across six translation directions with multiple controlled conditions.
4. The analysis is interesting and can be insightful for future work.

**Weaknesses:**

1. All experiments are limited to MT, leaving unclear how similar self-bias mechanisms apply to other generative or reasoning tasks.
2.  Each language direction includes only 200 instances, which limits statistical robustness and the strength of causal claims. Also, there is no significance analysis in the results.
3. The evaluated systems are all closed-source (Gemini, GPT-4.1, Claude), where open LLMs should be considered.

**Questions:**

Please see above.

---

> ### Author Response · Authors · 2025-11-25
> **For W1**
>
> Thank you for your constructive feedback!
> W1: Our goal was to work with a realistic setup that covers an important task where LLM-as-a-benchmark would be most relevant. Low-resource MT is currently a primary task where 'LLM-as-a-benchmark' is not just an option, but a necessity. Unlike high-resource tasks (e.g., English summarization or math reasoning) where human-curated test sets are plenty, low-resource languages lack ground-truth data. Consequently, the field is aggressively moving toward fully synthetic pipelines, we use LLMs to generate the training data, the test sets, and perform the evaluation. Therefore, validating self-bias in this specific pipeline is urgent. And as shown in our work, if the 'judge' and the 'generator' are the same model in a low-resource xx-en setting, we risk creating a closed-loop echo chamber where models are rewarded for their own hallucinations or specific 'translationese' styles, rather than genuine translation quality. However, our work is extensible to other tasks since each component such as LLM-as-a-testset, LLM-as-an-evaluator can be reformulated into other tasks. We leave further extension to other tasks to future work. In the paper, we constrained our claim to self-bias in translation benchmarks only.

---

> > ### Comment · Reviewer_cz1b · 2025-11-27
> >
> > Thanks to the authors for their reply.
> >
> > To my knowledge, there has been lots of work in low-resource summarization or cross-language summarization, such as CrossSum and ConvSumX, which perfectly suits the motivation of this work, as well as the explanation in their reply.

---

> > > ### Author Response · Authors · 2025-11-29
> > > **Add additional experiments of self-bias at Chatbot Arena task**
> > >
> > > Thank you for your suggestion! We further extend our self-bias experiments into Chatbot Arena task.
> > >
> > > Adopting the prompts of (https://arxiv.org/abs/2504.01001, **We leverage existing prompt sets that have demonstrated effective benchmark creation capabilities, rather than devising our own for this study**), we employ an LLM-as-a-benchmark framework to construct the Chatbot Arena task (https://arxiv.org/abs/2403.04132). We detail the prompts for LLM-as-a-test, LLM-as-an-evaluator, and LLM-as-a-benchmark in Appendix B from line 755 to 853; this diverse prompt set has demonstrated a strong correlation with human-generated benchmarks (https://arxiv.org/abs/2504.01001).
> > >
> > > Consistent with our previous experimental setup, we benchmark two distinct settings: high-resource language prompts (English and Korean) versus low-resource language prompts (Bemba and Aymara). As shown in Table 14, the results align closely with our findings in the translation task. Gemini-2.5-pro and Claude-Opus-4 exhibit minimal self-bias in English and Korean prompts but show a marked increase in self-bias when prompted in Bemba and Aymara. Specifically, the diagonal entries for Bemba and Aymara are consistently below 0, indicating significant self-bias.
> > >
> > > For high-resource languages (averaged), Gemini, GPT-4.1, and Claude show self-bias scores of 0.025, -0.207, and -0.017, respectively. In contrast, for low-resource languages, these scores shift to -0.301, -0.316, and -0.125. This trajectory highlights a sharp increase in self-bias when transitioning from high- to low-resource languages.
> > >
> > > I include table of English and Bemba results here. For all four table results, you can refer to our revised paper (from line 946 to 960).
> > >
> > > | Generater\LLM-as-a-benchmark (English) | Gemini | GPT4.1 | Claude |
> > > | :--- | :---: | :---: | :---: |
> > > | **Gemini** | -0.018 | -0.122 | 0.140 |
> > > | **GPT4.1** | 0.035 | **-0.205** | 0.170 |
> > > | **Claude** | 0.103 | -0.107 | 0.005 |
> > >
> > > | Generater\LLM-as-a-benchmark (Bemba) | Gemini | GPT4.1 | Claude |
> > > | :--- | :---: | :---: | :---: |
> > > | **Gemini** | **-0.385** | -0.010 | 0.396 |
> > > | **GPT4.1** | 0.052 | **-0.218** | 0.167 |
> > > | **Claude** | 0.334 | -0.154 | **-0.180** |

---

> ### Author Response · Authors · 2025-11-25
> **For W2**
>
> W2: To demonstrate the statistical significance of our sample size, we conducted a bootstrapping experiment on the Bemba-to-English translation task. We first replicated the 3x3 model ranking table using the LLM-as-a-Testset and LLM-as-an-evaluator setup for Gemini, Claude, and GPT4.1. Then, we performed bootstrapping with a sample size of N=200 and repeated 10,000 times to calculate confidence intervals. As shown in the tables below, the diagonal entries—representing a model's ranking of itself—exhibit no confidence interval overlap with the other entries in the same column. This result ensures that each model ranks itself higher than the other two with statistical significance. Based on this confidence interval analysis and budget constraints, we chose a sample size of 200 for all our experiments and will include this discussion to further justify our sample size choices.
>
> | Model | Gemini | GPT4.1 | Claude |
> | :--- | :---: | :---: | :---: |
> | **Gemini** | 1.00 ± 0.03 | 1.52 ± 0.09 | 1.70 ± 0.10 |
> | **GPT4.1** | 1.21 ± 0.07 | 1.18 ± 0.05 | 1.54 ± 0.09 |
> | **Claude** | 1.64 ± 0.11 | 1.93 ± 0.11 | 1.17 ± 0.06 |
>
> | Model | Gemini | GPT4.1 | Claude |
> | :--- | :---: | :---: | :---: |
> | **Gemini** | [0.97, 1.02] | [1.43, 1.61] | [1.59, 1.80] |
> | **GPT4.1** | [1.15, 1.28] | [1.12, 1.23] | [1.45, 1.64] |
> | **Claude** | [1.52, 1.75] | [1.82, 2.04] | [1.11, 1.23] |

---

> ### Author Response · Authors · 2025-11-25
> **For W3**
>
> Our model selection was strictly determined by the requirements of the low-resource Translation task. While we considered available open-source models, such as Gemma3-27B, Mistral-large-2411, and Qwen3, which officially support the target languages, our preliminary analysis and prior studies revealed a significant performance gap. These models are currently not suited for the LLM-as-a-benchmark pipeline in this setting, exhibiting a high degeneration rate when acting as an LLM-as-a-testset (found in source texts). Furthermore, they have a proven evaluation gap compared to the closed-source cluster (Gemini-2.5-pro, GPT4.1, and Claude-Opus-4) when acting as an LLM-as-an-evaluator, as demonstrated by prior work (https://www2.statmt.org/wmt25/pdf/2025.wmt-1.23.pdf).
>
> Since our bias formulation leverages the average estimation of the other two models as the proxy for true performance, selecting models with roughly the same evaluation capability was a scientific necessity to ensure the validity of our results. We leave further extension to other tasks and models to future work where open-source models have parity with closed-source models in both the LLM-as-a-test and LLM-as-an-evaluator roles to further enhance this study. We will include this discussion in the paper to further clarify our model and task choices.

---

### Official Review · Reviewer_c4v9 · 2025-11-01

**Soundness:** 3
**Presentation:** 3
**Contribution:** 1
**Rating:** 2
**Confidence:** 5

**Summary:**

The paper shows that using LLMs to generate training data (even just inputs) or to act as a judge, biases any evaluation that also uses the same LLM as a participating system. The paper is well written but overall this is such an obvious problem that it does not really require such detailed investigation. There are not really any new insights presented.

**Strengths:**

The experiments are well designed and executed.

The paper is well written.

The analysis of lack of variety of generated sources is interesting.

**Weaknesses:**

The overall point that LLMs are biases when used as both data generator/judge and participant is a very obvious problem. Anything a model produces will lie well within its distribution, and therefore it is likely to cover it well.

The particular setup to use LLMs to generate _source_ sentences for translation is a bit - there is typically not a lack of monolingual sentences to act as source, the problem is the lack of accurate reference translations. Of course, generating reference translations with an LLM and then treating them as gold standard is even more problematic (and it is the reason why at WMT shared tasks the professional translators are prohibited from using any machine translation tool).

**Questions:**

n/a

---

> ### Author Response · Authors · 2025-11-25
> **Regarding to weakness**
>
> Thank you for taking time for the feedback!
>
> We respectfully offer a different perspective regarding the reviewer’s point that self-bias is an "obvious" problem. While the intuition that a model prefers its own distribution is valid, the current research landscape—specifically the rapid adoption of frameworks like LLM-as-a-benchmark (https://arxiv.org/abs/2504.01001, https://arxiv.org/abs/2310.16049),  LLM-as-a-testset (https://arxiv.org/abs/2406.11939), and LLM-as-an-evaluator (https://arxiv.org/abs/2306.05685)—often operates without quantifying this bias. Prior works have largely either ignored self-bias entirely or assumed its presence without decomposing exactly how it affects fair evaluation (https://arxiv.org/abs/2504.03846). We argue that in the modern LLM era, where generation and evaluation utilize completely different prompt sets, the models effectively act as separate entities. Consequently, the connection between the two is non-trivial.
>
> Our work is the first to systematically decompose the LLM-as-a-Benchmark framework to isolate these individual effects. Crucially, we find that self-bias is not a blanket phenomenon. For instance, while the framework displays negligible self-bias when generating English source texts, it becomes a significant bottleneck for low- and medium-resource languages. We dive deep and show that this disparity stems from the model's limited generation capabilities in medium/low resource languages, which constrain both output diversity and quality. Consequently, our findings challenge the assumption that self-bias is uniform and offer vital guidelines for future research in this emerging direction.
>
> Regarding the practical perspective of our setup, we believe this approach is highly valuable for low and medium-resource languages. While high-resource languages may have abundant monolingual data, finding clean, high-quality benchmark data for lower-resource languages remains a challenge. Our method extends the LLM-as-a-Benchmark paradigm to these settings where traditional benchmarks fail to scale. Furthermore, we argue source texts + strong reference free metrics (on par with reference-based metrics, shown in WMT25, https://www2.statmt.org/wmt25/pdf/2025.wmt-1.23.pdf) is a strong alternative compared to have additional dependencies in generating reference texts which could amplify bias. Related to this point, we have a surprising and significant finding in the paper: self-bias can emerge even when the LLM is used strictly to generate source texts and the evaluation relies on external metrics (such as MetricX). In addition, we prioritized the source-only setting to ensure our findings generalize to tasks where reference texts are inherently unavailable (more general case nowdays), where LLM-as-an-evaluators only see prompt and response without reference texts.

---

> > ### Comment · Reviewer_c4v9 · 2025-11-26
> >
> > I follow your argument for the premise of the study. I still have a different opinion on that, and I'll leave it to the meta reviewer to balance these different points of view.
> >
> > As stated in the review, the experimental work is strong.

---

### Official Review · Reviewer_BRFz · 2025-11-01

**Soundness:** 2
**Presentation:** 3
**Contribution:** 2
**Rating:** 4
**Confidence:** 3

**Summary:**

This paper studies self-bias in LLM-generated translation benchmarks. It first decomposes the concept of LLM-as-a-benchmark into two components: LLM-as-a-testset and LLM-as-an-evaluator. The observed bias arises from the interaction between these two roles and is more pronounced in low-resource-to-English translation tasks. The paper further demonstrates that limited diversity in the source texts contributes to self-bias, while increasing source-text diversity can mitigate it. Despite this limitation, LLM-as-a-benchmark remains a valuable approach for ranking weaker models and for evaluating translations.

**Strengths:**

1. The paper conceptualizes LLM-as-a-benchmark as consisting of two distinct components, LLM-as-a-testset and LLM-as-an-evaluator, and provides dedicated analyses benefiting from this decomposition.
2. LLM-as-a-benchmark remains a valuable approach for ranking weaker models and for evaluating translations.

**Weaknesses:**

The testset generation process in the LLM-as-a-benchmark framework is not sufficiently well designed. A well-constructed benchmark should incorporate both diversity control and quality control [1] [2] [3]. The conclusion that “generating more diverse source texts can mitigate self-bias” arises precisely because the LLM-as-a-benchmark approach adopted in the paper lacks diversity control, which leads to unnecessary self-bias. Similarly, the degeneration issue observed in the generated source texts (see Figure 1) results from the absence of quality control. Consequently, the self-bias observed in a simply generated testset is analyzed within an overly narrow and unrealistic context, which undermines the persuasiveness of the argument. Therefore, it is helpful for testset generation to incorporate appropriate diversity control and quality control, so that analyses of self-bias conducted on such datasets can be sufficiently convincing.

References

[1] https://lmsys.org/blog/2024-04-19-arena-hard

[2] Li, Tianle, et al. "From crowdsourced data to high-quality benchmarks: Arena-hard and benchbuilder pipeline." arXiv preprint arXiv:2406.11939 (2024).

[3] Lin, Bill Yuchen, et al. "WildBench: Benchmarking LLMs with Challenging Tasks from Real Users in the Wild." The Thirteenth International Conference on Learning Representations.

**Questions:**

1. Since self-bias is sensitive to the diversity and quality of the benchmark, it is helpful that the diversity and quality of the benchmark are presented in detail.
2. Moreover, studying self-bias across benchmarks with varying levels of diversity and quality provides more informative insights than studying self-bias on a simply generated benchmark.

---

> ### Author Response · Authors · 2025-11-25
> **For W1**
>
> We appreciate the helpful feedback and fully agree that benchmark diversity and quality are important factors to self-bias (This is one important finding in our paper). In our analysis, we found that the models' intrinsic struggle to produce diverse, high-quality text in low-resource languages is actually a primary mechanism driving the self-bias we observed. To clarify, our test set generation did incorporate explicit diversity controls by leveraging the framework established by [https://arxiv.org/abs/2504.01001], which randomizes topics, styles, and lengths for each prompt. Because prior work demonstrates that this prompt set achieves high correlation with human ratings, the homogeneity in our results reflects model limitations in these languages rather than a deficiency in prompt engineering—a conclusion supported by the fact that English-to-XX direction exhibited significantly less self-bias.
>
> Following your suggestion to isolate the variable of quality, we conducted an additional study by filtering the test set. Given the difficulty of direct quality measurement for Bemba and Aymara, we employed MetricX QE [https://arxiv.org/abs/2410.03983], a state-of-the-art metric where scores closer to 0 indicate higher quality within a range of -25 to 0. We selected the top 50 and lowest 50 source sentences based on their alignment with English pairs. Even among the top 50 high-quality Bemba sentences—where a score of -1 corresponds roughly to one minor error—the models achieved strong average MetricX QE scores of -0.78 (Gemini), -1.4 (GPT4.1), and -1.5 (Claude). Despite this high quality source texts, these models still displayed high self-bias on the Bemba-to-English task. These results confirm that while quality control mitigates the magnitude of the bias, it does not eliminate it, supporting our conclusion that low quality and diversity are contributing factors to self-bias.  However, the whole story of self-bias requires further research to fully understand.
>
> First Table is the average of second and third
>
> | XX->En (Self-bias) | Gemini | GPT4.1 | Claude |
> | :--- | :--- | :--- | :--- |
> | Low quality (50) | -0.47 | -0.37 | -0.76 |
> | Random (50) | -0.44 | -0.22 | -0.56  |
> | High quality (50) | -0.21 | -0.12 | -0.57 |
>
> | Bemba ->En (Self-bias) | Gemini | GPT4.1 | Claude |
> | :--- | :--- | :--- | :--- |
> | Low quality (50) | -0.72 | -0.31 | -0.79 |
> | Random (50) | -0.66 | -0.14 | -0.52 |
> | High quality (50) | -0.53 | -0.17 | -0.53 |
>
> | Aymara ->En (Self-bias) | Gemini | GPT4.1 | Claude |
> | :--- | :--- | :--- | :--- |
> | Low quality (50) | -0.22 | -0.42 | -0.73 |
> | Random (50) | -0.21 | -0.30 | -0.60 |
> | High quality (50) | 0.11 | -0.06 | -0.60  |

---

### Meta-Review · Area_Chair_SG5a · 2025-12-14

**Summary:**

This paper investigates a critical limitation in the emerging paradigm of automated benchmark creation using large language models (LLMs), where LLMs are used both to generate test sets and to evaluate model performance, especially for translation tasks. While LLM-generated benchmarks offer a scalable alternative to expensive human-curated datasets, the authors discover that these benchmarks systematically favor the model that created them, a phenomenon they term self-bias in translation evaluation.

The study makes three main contributions. First, it shows that self-bias arises from two additive sources: the test data generation component (“LLM-as-a-testset”) and the evaluation mechanism (“LLM-as-an-evaluator”); when combined, these amplify bias significantly. Second, the authors find that bias magnitude varies with the model’s generation capability in the source language: bias is more pronounced in into-English (XX→En) translation directions than in out-of-English (En→XX) tasks, likely because models generate source texts that favor their own translation strengths. Third, they identify low diversity in the generated source texts as a contributing factor to self-bias and show that increasing source‐text diversity can help mitigate bias.

Overall, the paper highlights that LLM-generated benchmarks may overestimate a model’s ability, especially in low-resource language settings, and that careful design is needed to ensure fair, objective evaluation.

**Reviewer Concerns:**

Concern 1. The benchmark should incorporate both diversity control and quality control. The conclusion that “generating more diverse source texts can mitigate self-bias” arises precisely because the LLM-as-a-benchmark approach adopted in the paper lacks diversity control, which leads to unnecessary self-bias.
Concern 2. The overall point that LLMs are biases when used as both data generator/judge and participant is a very obvious problem. Anything a model produces will lie well within its distribution, and therefore it is likely to cover it well.
Concern 3. Experiments are limited to the MT, leaving unclear how similar self-bias mechanisms apply to other generative or reasoning tasks and the evaluated systems are all closed-source (Gemini, GPT-4.1, Claude), where open LLMs (e.g., open source multilinguality focused LLMs) should be considered.

I think the Concern 1. & Concern 3. are partially addressed by the author's response while noting that the divergence still exist between reviewers and the author (espcially for Concern 1 & 2. two reviewers question the rationality of the basic experiment setting of the paper). As for Concern 3., though authors clarify that the open-source models are not applicable for their low-capacity, the author's concern on the sensitivity of ranking remains existing.

**Reviewer Scores:**

- Reviewer BRFz are likely to maintain the score (4);
- Reviewer c4v9 explicitly stated he maintain the score (2);
- The rest two reviewers would probably maintain the score (6 & 8).

---

### Decision · Program_Chairs · 2026-01-26

Reject